# PolygoNet: Leveraging Simplified Polygonal Data for Effective Image Classification

## Abstract

Deep learning models have achieved significant success in various image-related tasks. However, they often encounter challenges related to computational complexity and overfitting. In this paper, we propose an approach that leverages efficient polygonal representations of input images by utilizing either dominant points or coordinates of contours. Our method transforms input images into polygonal forms using one of these techniques, which are then employed to train deep neural networks. This representation offers a concise and flexible depiction of images. By converting images into either dominant points or contour coordinates, we substantially reduce the computational burden associated with processing large image datasets. This reduction not only accelerates the training process but also conserves computational resources, rendering our approach suitable for real-time applications and resource-constrained environments. Additionally, these representations facilitate improved generalization of the trained models. Both dominant points and contour coordinates inherently capture essential features of the input images while filtering out noise and irrelevant details, providing an inherent regularization effect that mitigates overfitting. Our approach results in lightweight models that can be efficiently deployed on edge devices, making it highly applicable for scenarios with limited computational resources. Despite the reduced complexity, our method achieve performance comparable to state-of-the-art methods that use full images as input. We validate our approach through extensive experiments on benchmark datasets, demonstrating its effectiveness in reducing computation, preventing overfitting, and enabling deployment on edge computing platforms. Overall, this work presents a methodology in image processing that leverages polygonal representations through either dominant points or contour coordinates to streamline computations, mitigate overfitting, and produce lightweight models suitable for edge computing. These findings indicate that this approach holds significant potential for advancing the field of deep learning by enabling efficient, accurate, and scalable solutions in real-world applications. The code for the experiments of the paper are provided at
https://anonymous.4open.science/r/PolygoNet-7374

## 1 Introduction

Image classification remains a cornerstone of computer vision, with applications spanning from autonomous vehicles to medical diagnostics. The increasing demand for real-time analysis on resource-constrained platforms necessitates efficient data representation and processing methods. Traditional approaches that rely on raw pixel data often encounter substantial computational costs and memory requirements, challenges that are exacerbated when handling high-resolution images. Handling high-resolution imagery increases data volume and computational load, making conventional pixel-based methods less practical for real-time applications. This situation highlights the need for techniques that reduce data complexity while retaining the essential features necessary for accurate classification. To address these challenges, we propose an approach that utilizes either dominant points or the coordinates of contours extracted from image contours as a compact and effective representation for classification tasks. This methodology departs from traditional pixel-level analysis by focusing on geometrically salient features captured through image contours, implementing an implicit form of image classification. Specifically, our approach can employ either the raw coordi-

nates of contours extracted from the shapes within images or use the Modified Adaptive Tangential Cover (MATC) algorithm Ngo et al. (2017); Ngo (2019) to extract dominant points that succinctly capture the essential shape information with fewer points.

The use of either contour coordinates or dominant points significantly reduces data dimensionality while preserving critical geometric attributes essential for effective classification. Extracting the full contour coordinates provides a detailed representation of an object's shape, while using dominant points via MATC offers a more concise representation by identifying key structural points, thus reducing the number of data points required. This flexibility allows the model to process data more efficiently, reducing computational overhead and making it suitable for devices with limited processing capabilities, such as CPUs and edge computing platforms. Importantly, despite the reduced data representation, our method achieves classification performance that is practically comparable to state-of-the-art methods that use full images as input. By concentrating on the structural essence of images, the approach enhances the ability to generalize from minimal data and diminishes the influence of background noise or irrelevant variations.

This methodology aligns with cognitive processes observed in human visual perception, where recognition is often based on key structural features rather than exhaustive pixel-by-pixel analysis Biederman (1987); Koffka (2013). Mimicking this aspect may improve computational efficiency and potentially increase classification accuracy by emulating how humans perceive and categorize visual information.

In summary, the proposed method addresses the challenges of high-resolution image classification by employing either contour coordinates or dominant point extraction through MATC to achieve a compact yet informative data representation. This approach reduces computational requirements by lowering data dimensionality, enabling image classification with fewer resources and on devices with limited processing capabilities. Crucially, it maintains classification performance comparable to state-of-the-art methods using full images, thereby improving the speed and efficiency of real-time image classification tasks without sacrificing accuracy. This contributes to advancements in edge computing and mobile AI applications, where resource constraints are a significant concern.

## 2 RELATED WORK

**Image Classification.** Image classification is a fundamental task in computer vision, aiming to assign predefined labels to images. Deep learning architectures for this task have predominantly been based on convolutional neural networks (CNNs). Since the breakthrough of AlexNet (Krizhevsky et al., 2012), CNNs have become the standard for image recognition, with notable architectures such as VGG (Simonyan & Zisserman, 2014), Inception (Szegedy et al., 2015), ResNet (He et al., 2016), and EfficientNet (Tan & Le, 2019) advancing the field. Concurrently, the success of self-attention mechanisms in natural language processing, particularly with Transformers (Vaswani et al., 2017; Devlin et al., 2018; Brown et al., 2020), has inspired their integration into computer vision models (Wang et al., 2018; Bello et al., 2019; Srinivas et al., 2021; Shen et al., 2021). A significant development is the Vision Transformer (ViT) (Dosovitskiy et al., 2020), which demonstrates that pure Transformer architectures can achieve competitive performance on image classification tasks.

**Shape and Contour Analysis.** Early methods for contour classification relied on handcrafted features to represent shapes. Techniques like Shape Context (Belongie et al., 2002) and Fourier Descriptors (Kuhl & Giardina, 1982) capture global and local contour information, focusing on extracting discriminative features from object boundaries. These approaches laid the groundwork for contour representation and classification. With the advent of deep learning, CNNs have been adapted to process contour information (Baker et al., 2018; 2020), showing improved performance in tasks such as handwritten digit recognition and object classification based on boundary information. These models leverage the hierarchical feature extraction capabilities of deep networks for effective contour representation.

**Self-Attention Mechanisms.** Self-attention is the core component of Transformer architectures, allowing models to learn dependencies across input tokens without the locality constraints of CNNs. Introduced by Bahdanau et al. (2014) for neural machine translation, attention mechanisms enable models to weigh the importance of different parts of the input sequence, capturing long-range dependencies more effectively. This capability has been successfully applied to various natural language

processing tasks, including image captioning (Xu et al., 2015) and sentiment analysis. In computer vision, self-attention mechanisms have been incorporated to capture global context. Wang et al. (2018) introduced non-local neural networks that compute responses at a position as a weighted sum of features at all positions, enabling the network to model global information. The Vision Transformer (ViT) (Dosovitskiy et al., 2020) further adapted the Transformer architecture to vision tasks by treating image patches as tokens, leveraging self-attention to model interactions across the entire image.

**Combining CNNs with Self-Attention.** The integration of CNNs with self-attention mechanisms has garnered significant interest due to its potential to enhance performance across various domains. This hybrid approach has improved image classification by incorporating self-attention into CNN feature maps (Bello et al., 2019), and has been effectively applied to object detection (Hu et al., 2018; Carion et al., 2020) and video processing (Wang et al., 2018; Sun et al., 2019). The synergy between CNNs and self-attention also advances unsupervised object discovery (Locatello et al., 2020) and facilitates multimodal tasks that bridge text and vision (Chen et al., 2020; Lu et al., 2019; Li et al., 2019).

Our work leverages this combination of self-attention mechanisms with convolutional architectures. Self-attention efficiently integrates features that are spatially distant in the input representation and naturally handles variable input sizes. As detailed in the methodology section, encoding shapes with dominant points results in inputs of variable length, since complex shapes require more points to be effectively encoded than simpler ones.

## 3 METHOD

### 3.1 DATA PREPROCESSING

Data preprocessing is a critical component of our methodology, as the proposed architecture operates on coordinate inputs rather than raw pixel data. Specifically, we can directly use either the contours extracted from the shapes within the images or the dominant points derived from these contours. Figures 1 and 2 illustrate the preprocessing steps and the generation of cordinates points, highlighting the two distinct pipelines in our methodology. The first pipeline involves directly extracting the contour coordinates from the shapes within the images, providing a detailed representation of the object's outline. The second pipeline applies the Modified Adaptive Tangential Cover (MATC) algorithm to compute dominant points, resulting in a more concise representation by capturing key structural features. The number of points obtained in each method varies depending on the approach used and the complexity of the shape.

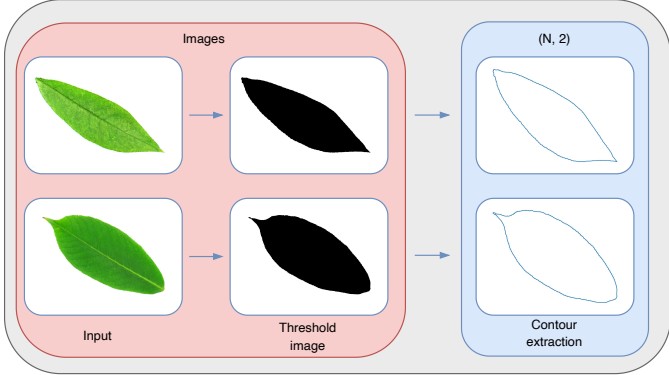

Figure 1: The first step in our shape encoding process involves applying thresholding to the image to segment the object from the background. This is followed by extracting the contours using one of the methods detailed in Section 3.1.1. The number of contour points obtained varies depending on the extraction method used and the complexity of the shape.

### 3.1.1 CONTOUR EXTRACTION

In our approach, contours are extracted from images using various contour approximation techniques to generate coordinate-based representations of object shapes. Specifically, we employ the following methods:

- **No Approximation (None)** (Bradski, 2000): This method retains all contour points without any simplification, ensuring that each pair of consecutive points remains connected through horizontal, vertical, or diagonal neighbor relations. This means that for any consecutive pairs $(x_1, y_1)$ and $(x_2, y_2)$, the condition $\max(|x_1 - x_2|, |y_1 - y_2|) = 1$ holds true, guaranteeing strict connectivity along the contour.

- **Simple Approximation** (Suzuki et al., 1985): This method simplifies contours by removing all redundant points that form horizontal, vertical, or diagonal straight-line segments, retaining only the starting and ending points of these segments. This reduces the number of points while preserving the essential shape characteristics.

- **TC89-L1 Approximation**: Utilizing an algorithm based on the approach proposed by Teh & Chin (1989), this method simplifies contours by approximating their shape with polygonal segments. The TC89-L1 approximation applies an L1 (Manhattan distance) measure, which favors simpler contours while maintaining good geometric fidelity.

- **TC89-KCOS Approximation**: Also based on the method proposed by Teh & Chin (1989), this approximation uses a cosine distance (KCOS) measure. It provides a smoother polygonal approximation of contours, making it suitable for more complex shapes by better preserving curvatures and geometric details.

By applying these contour approximation techniques, we can control the level of detail in the contour representations, balancing between data compactness and shape fidelity. This allows us to generate input data that is both efficient for processing and rich in essential geometric features necessary for accurate classification.

### 3.1.2 MODIFIED ADAPTIVE TANGENTIAL COVER (MATC) APPROACH

The *Modified Adaptive Tangential Cover (MATC)* approach plays a significant role in simplifying data preprocessing within our methodology, particularly in the precise approximation of contours, as demonstrated by Ngo (2019). MATC is founded on the principles of fuzzy segments and tangential cover, defined as a sequence of fuzzy segments with variable thickness $\nu$. According to Kerautret et al. (2012), this thickness dynamically adjusts in response to local noise levels present along a digital curve.

MATC proves particularly effective due to its robustness against noise and imperfections commonly observed in digital contours. By effectively addressing these anomalies, MATC preserves the integrity of the approximated contours, ensuring that the data used in subsequent processing steps or analytical applications maintain high fidelity to the original geometric characteristics. Dominant points, which are essential for representing the geometric properties of contours, are identified within the smallest common regions formed by successive fuzzy segments. These points are characterized by their minimal curvature, facilitating their detection through straightforward angle measurements. The steps involved in computing dominant points using the MATC approach are as follows:

1. **Digital Contour Extraction**: This step is equivalent to the previous stage 3.1.1, where the goal is to extract object contours from a digital image. These contours are represented as numerical curves, where the points have integer coordinates $(x, y)$.

2. **Computation of Adaptive Tangential Covering**: This step involves applying a tangential covering to the extracted numerical curves. The process consists of dividing the curve into a sequence of blurred segments with varying thicknesses, which change according to the level of local noise detected along the curve. The segment thickness is adjusted using a local noise estimator called "meaningful thickness."

3. **Dominant Points Identification**: Dominant points are localized in the smallest common areas created by successive blurred segments. At each candidate point, an angle measurement (pseudo-curvature) is performed to identify the point with the smallest angle within

this area. This point is identified as a local maximum curvature point, thus a dominant point.

4. **Polygonal Simplification**: After identifying the dominant points, the contour is simplified to obtain a polygonal approximation of the curve. Dominant points that are too close to each other are eliminated to reduce complexity and improve efficiency while maintaining the geometric fidelity of the original contour.

5. **Optimization**: The simplification process includes an evaluation of the quality of the generated polygon using criteria such as the sum of squared errors (ISSE) and the compression ratio (CR). A score is assigned to each point based on its importance to the curve, and points are eliminated until an optimal balance between approximation fidelity and data compression is achieved.

The Modified Adaptive Tangential Cover (MATC) approach is designed to provide a robust and adaptive polygonal approximation of digital contours by accounting for local noise variations and preserving essential geometric characteristics. This methodology enables the efficient representation of complex curves with a reduced number of points, which not only simplifies analytical processing but also decreases the number of parameters required for model training. Consequently, MATC enhances both the efficiency and overall performance of the classification model by facilitating streamlined data processing and minimizing computational overhead.

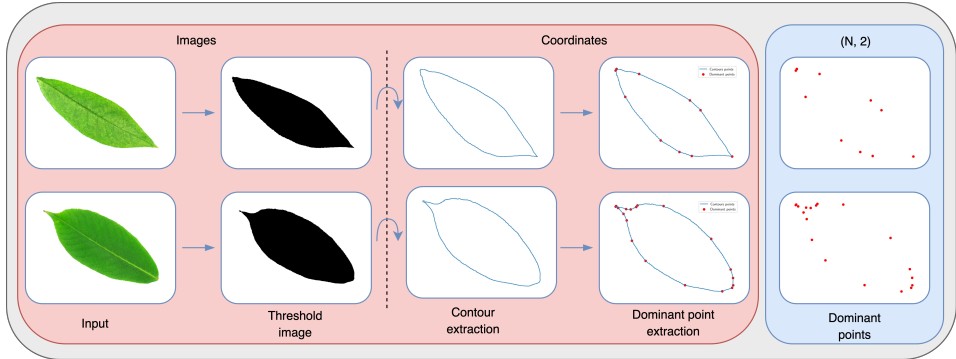

Figure 2: The initial step in encoding a shape begins with applying thresholding to the image, followed by contour extraction, and finally applying the Modified Adaptive Tangential Cover (MATC) algorithm to compute the dominant points. The number of dominant points is variable and depends on the complexity of the shape.

Let $I \in \mathbb{R}^{H \times W \times C}$ denote an input image, where $H$, $W$, and $C$ represent the height, width, and number of color channels, respectively. The extraction process of a set of $N$ dominant points, $D$, begins with converting the RGB image to grayscale. This conversion simplifies the data while preserving essential visual information. Following this, thresholding is applied to the grayscale image to generate a binary image. Additionally, filtering techniques are employed to eliminate noise and enhance the clarity of the shapes. Contours, $C = \{c_i \in \mathbb{R}^2\}$, are then extracted from the processed image. Subsequently, the Modified Adaptive Tangential Cover (MATC) algorithm Ngo (2019) is applied to these contours to identify and extract the dominant points, $D$. The number and positions of these dominant points can vary significantly between images, reflecting the unique characteristics and structural variations inherent in each image. These dominant points, $D$, are represented as an $N \times 2$ matrix, where each row corresponds to the $(x, y)$ coordinates of a dominant point in the image plane.

The pseudo-code 1 outlines the various steps employed during the data preparation process using MATC.

### 3.2 Networks

**Baseline.** We adopted the ResNet architecture He et al. (2016) as our baseline CNN, utilizing RGB images. ResNet was trained on the same dataset used for extracting dominant points, ensuring

---

**Algorithm 1** Extraction of Dominant Points from Image

---

**Require:** Input image $I \in \mathbb{R}^{H \times W \times C}$         ▷ e.g. Flavia Image size: $(1600 \times 1200 \times 3)$
**Ensure:** Matrix of dominant points $D$ with dimensions $N \times 2$     ▷ Avg dimension of D: $(60 \times 2)$
    $\mathcal{I}_g \leftarrow \text{Grayscale}(I)$                                  ▷ Converts $I$ to grayscale
    $\mathcal{I}_b \leftarrow \text{Threshold}(I_g)$     ▷ Thresholds the grayscale image to produce a binary mask of the shape
    $\mathcal{C} \leftarrow \text{ExtractContours}(I_b)$                         ▷ Extract contour points from $I_b$
    $D \leftarrow \text{ApplyMATC}(\mathcal{C})$                  ▷ Apply Modified Adaptive Tangential Cover on $\mathcal{C}$
    **return** $D$                                     ▷ Return the matrix of dominant points

---

consistent metrics and a fair comparison. We evaluated ResNet-18, ResNet-34, and ResNet-50, reporting the best-performing variant. Although Vision Transformers (ViTs) Dosovitskiy et al. (2020) demonstrate strong performance, especially on large-scale datasets, we chose ResNet for its established architecture, ease of implementation, and lower computational demands. ResNet-50 is particularly effective in scenarios with limited data, enabling a fair assessment of our proposed approach. By processing 3-channel RGB images, ResNet leverages rich color information to capture detailed variations, textures, and contextual cues essential for distinguishing visually similar objects.

**PolygoNet.** To address the challenge of processing variable-length coordinates extracted from original input images, the architecture developed in this paper introduces an adaptation of the self-attention mechanism, inspired by the works of Vaswani et al. (2017); Dosovitskiy et al. (2020) on Transformer models. This methodology enables our model to dynamically adapt to the input space, efficiently handling point sets regardless of their size. By leveraging the capabilities of self-attention, the model can assign appropriate weights to each point, thereby capturing the complex geometric nuances specific to the dataset. The model computes attention scores using the normalized dot product of queries, keys, and values, facilitating a weighted assessment of the importance of each input token relative to others. This approach ensures that the extracted features faithfully reflect the essential geometric properties of the shapes, accurately capturing their structures, forms, and inter-point relationships. Consequently, critical information necessary for precise and thorough shape analysis is preserved and emphasized by the model. The incorporation of 1D convolutional blocks further enhances feature extraction, enabling the model to detect complex geometric patterns in the coordinates point data. The architecture is illustrated in Figure 3. Specifically, the architecture integrates Multi-Head Self-Attention (MSA) layers as utilized in Dosovitskiy et al. (2020), alongside Conv1D blocks, thereby enhancing its ability to process geometric data effectively. Each block is preceded by a normalization layer, which standardizes the data to facilitate more stable and efficient learning Ioffe & Szegedy (2015). In the architecture depicted in Figure 3, $f_\theta$ represents the Conv1D blocks, with each layer followed by a normalization layer and a ReLU activation function. The MLP head consists of a simple linear layer with the number of classes as its parameter. The use of 1D convolutional (Conv1D) layers is particularly effective in this context due to their capacity for capturing local dependencies and patterns along the sequence of points and for computational efficiency, thereby augmenting the attention mechanism's global perspective with localized feature extraction. This sequential application of self-attention followed by Conv1D processing allows our model to enhance model's performance by effectively capturing both global dependencies and local patterns within the dominant point coordinates. The proposed method integrates global attention mechanisms with localized convolutional processing to effectively extract variable-length geometric features, addressing associated challenges with improved precision and robustness. Positional embeddings are incorporated with dominant points coordinates to preserve positional data. In the context of our approach, the positional embedding refers to the ordered sequence that defines the form and structure of the shapes, enabling the model to incorporate the sequential arrangement into its understanding and processing. There are several choices of positional embedding, our method uses 1D learnable positional embedding as a standard approach which is based on the sine and cosine function of different frequencies Vaswani et al. (2017). PolygoNet processes an input tensor of shape $(N, 2)$, where $N$ represents the number of points. The architecture begins with a custom attention mechanism to effectively capture relevant features from the input. It comprises five sequential 1D convolutional layers with increasing output channels: 64, 128, 256, 512, and 1024. Each convolutional layer is followed by batch normalization and a ReLU activation function to enhance feature learning and model stability. Specifically, the first layer includes an additional dropout layer with a dropout rate of $10\%$ to prevent overfitting. The network culminates in a classification head that

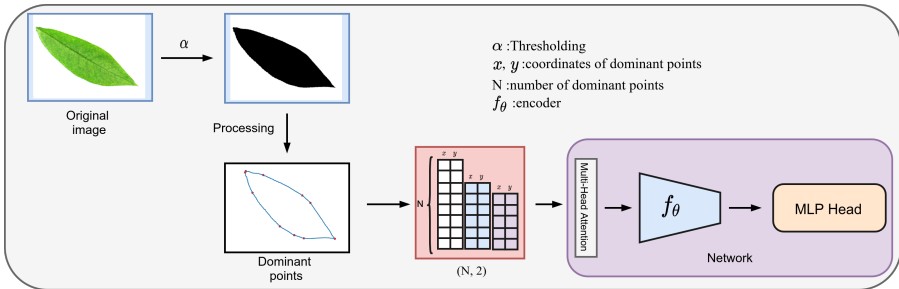

Figure 3: PolygoNet pipeline. The input colored image is converted to grayscale before being thresholded with Otsu. The dominant points are extracted using the MATC approach from the extracted contour. This variable size sequence of dominant points is then processed for classification by PolygoNet. Note that the complexity of the contour impacts the number of computed dominant points that will be processed by PolygoNet.

outputs predictions across the specified number of classes, resulting in an output tensor of shape $(num\_classes)$.

The integration leverages a standard approach using sine and cosine functions to provide unique positional encodings for each position, enabling the model to distinguish points based on their sequence positions. Specifically, each position *pos* is encoded with sine and cosine functions of varying frequencies to capture both absolute and relative positions. The positional encoding for a given position *pos* and dimension $i$ is defined as:

$$PE_{(pos,2i)} = \sin\left(\frac{pos}{10000^{2i/d}}\right) \quad \text{and} \quad PE_{(pos,2i+1)} = \cos\left(\frac{pos}{10000^{2i/d}}\right)$$

where $d$ is the dimensionality of the model.

By leveraging these positional encoding, our model can effectively retain the sequential and spatial relationships among the dominant points, enhancing its ability to capture the geometric the structure of the shapes.

## 4 EXPERIMENTS

In this section, we explore the usage of our proposed approach for image classification task. We show results on three different datasets.

**Datasets** To comprehensively evaluate our model's performance and robustness, we conducted experiments on three image classification datasets: **FashionMNIST** Xiao et al. (2017) consists of 70,000 grayscale images with a resolution of $28 \times 28$ pixels across 10 classes. **Flavia** Wu et al. (2007) includes 1,900 high-resolution leaf images ($1600 \times 1200$ pixels) spanning 32 classes, presenting subtle inter-class variations that challenge classification accuracy. **Folio** Munisami et al. (2015) contains 32 plant classes, each represented by 20 RGB images at a resolution of $4160 \times 3120$ pixels, featuring diverse lighting conditions and varying scales to simulate real-world imaging scenarios. These datasets were selected for their well-segmented objects against uniform backgrounds, facilitating effective contour extraction and enabling our pipeline to demonstrate consistent performance across diverse and challenging conditions.

**Implementation Details** All experiments employ the Adam optimizer (Kingma & Ba, 2014) with hyperparameters $\beta_1 = 0.9$, $\beta_2 = 0.999$, a learning rate of $10^{-5}$, and a weight decay of 0.0001. To enhance regularization, a dropout layer (Srivastava et al., 2014) with a dropout rate of $10\%$ is applied, effectively masking neurons during training to improve generalization. For data augmentation, the ResNet-50 architecture utilizes rotations and horizontal/vertical flips to increase training diversity and robustness. Similarly, PolygoNet, which processes coordinate inputs, applies analogous rotations and flips to the coordinate data to maintain consistency and enhance generalization across varied input representations. The ResNet-50 model is trained for 150 epochs, whereas PolygoNet undergoes 300 epochs to ensure comprehensive learning. Early stopping is implemented in all experiments to prevent overfitting, with the best validation performance recorded. Training is conducted on a single NVIDIA RTX 3090 GPU, and select experiments are also performed on a

CPU to demonstrate the approach's efficiency under different hardware constraints. For inference and processing time evaluations, the NVIDIA Jetson Orin Nano is utilized. This embedded system, featuring an Ampere-based GPU, supports complex inference tasks while maintaining a compact form factor and energy efficiency, making it ideal for real-time AI applications.

**Metrics** Across all experiments, we utilized two quantitative metrics to assess the quality and performance of the developed approach: Accuracy and F1-score.

**Evaluation Methodology** To demonstrate the generalization capabilities of our approach across different coordinate acquisition modalities, we evaluated two distinct pipelines: dominant point-based evaluation and contour point-based evaluation.

- **Evaluation on Contours**: In this evaluation, we extracted the contour coordinates from the input images using four methods outlined in 3.1.1. The processing time assessment therefore comprises the time taken for contour extraction and inference.

- **Evaluation on Dominant Points**: For this assessment, we employed the MATC method detailed in 3.1.2 to generate dominant points from the contours extracted from the input images. The processing time evaluation involves summing the durations of each component, specifically contour extraction, dominant point calculation, and inference.

## 5 RESULTS

**Evaluation on FashionMNIST** We evaluated our model on the standard FashionMNIST split, comprising 60,000 training and 10,000 test grayscale images of size $28 \times 28$ pixels across 10 classes. PolygoNet was trained with a batch size of 64, demonstrating robustness against overfitting and maintaining stability despite the extended training duration. Specifically, PolygoNet (DP) achieved an F1-score of 0.90 and an accuracy of 79%, while PolygoNet (Contours) improved to an F1-score of 0.91 and an accuracy of 83%, both with low computational complexity of approximately 8.5 million FLOPs. In contrast, ResNet-50 attained a higher accuracy of 90% and an F1-score of 0.93 but with a significantly greater computational cost of 80.38 million FLOPs.

**Evaluation on Flavia** We evaluated our model on the Flavia dataset, which comprises 1,900 high-resolution leaf images resized to $512 \times 512$ pixels for ResNet to accommodate GPU memory constraints. PolygoNet (DP) achieved an F1-score of 0.90 and an accuracy of 79%, while PolygoNet (Contours) improved the accuracy to 83%, both maintaining low computational costs of 8.67 million and 8.80 million FLOPs, respectively. In contrast, ResNet-50 attained a higher accuracy of 91% with an identical F1-score of 0.90 but at a significantly greater computational cost of 21.47 billion FLOPs.

**Evaluation on Folio** We evaluated our model on the **Folio** dataset, featuring diverse lighting conditions and varying scales. **PolygoNet (DP)** achieved an F1-score of 0.88 and an accuracy of 78% with a computational cost of 8.66 million FLOPs. **PolygoNet (Contours)** maintained the same F1-score of 0.88 while improving accuracy to 81%, incurring a slightly higher FLOPs count of 8.79 million. In contrast, **ResNet-50** attained a higher accuracy of 86% and an F1-score of 0.84 but with a significantly greater computational expense of 21.47 billion FLOPs.

As shown in Table 1, PolygoNet variants achieve competitive F1-scores and accuracies with significantly lower FLOPs compared to ResNet-50, underscoring their computational efficiency and effectiveness, highlighting its suitability for resource-constrained environments.

**Processing Time Evaluation** Table 2 presents the benchmark results for PolygoNet and ResNet-50 across three datasets (**FashionMNIST**, **Folio**, and **Flavia**) and two device configurations (GPU server and Jetson Orin). These results provide a comprehensive comparison of each pipeline's computational efficiency and practicality under different settings. Table 3 summarizes the processing time benchmarks for PolygoNet (with variant contours extractions methods) and ResNet-50 across three datasets (**FashionMNIST**, **Folio**, and **Flavia**) and two device configurations (GPU server and Jetson Orin). These results provide a comprehensive comparison of each pipeline's computational efficiency and practicality under different settings.

Table 1: Performance Comparison of Models Across Various Datasets

| Dataset | Method | F1-score ↑ | Accuracy ↑ | FLOPs ↓ |
|---|---|---|---|---|
| FashionMNIST | PolygoNet (DP) | 0.90 | 0.79 | **8.52 M** |
| | PolygoNet (Contours) | 0.91 | 0.83 | 8.65 M |
| | ResNet-50 | 0.93 | 0.90 | 80.38 M |
| Flavia | PolygoNet (DP) | 0.90 | 0.79 | **8.67 M** |
| | PolygoNet (Contours) | 0.90 | 0.83 | 8.80 M |
| | ResNet-50 | 0.90 | 0.91 | 21.47 G |
| Folio | PolygoNet (DP) | 0.88 | 0.78 | **8.66 M** |
| | PolygoNet (Contours) | 0.88 | 0.81 | 8.79 M |
| | ResNet-50 | 0.84 | 0.86 | 21.47 G |

Table 2: Benchmarking Processing Time of Two Pipelines on Three Datasets Across Two Configuration

| Dataset | Device | Pipeline | Contour Extract (ms) | MATC (ms) | Inference (ms) | Total Time (ms) |
|---|---|---|---|---|---|---|
| FashionMNIST (28 × 28) | Workstation | Our | 1.68 | 6.22 | 1.76 | **9.66** |
| | | ResNet-50 | - | - | 17.06 | 17.06 |
| | Edge Computing | Our | 2.28 | 54 | 6.15 | **62.43** |
| | | ResNet-50 | - | - | 116.25 | 116.25 |
| Flavia (1600 × 1200) | Workstation | Our | 13.80 | 125 | 1.51 | **140.31** |
| | | ResNet-50 | - | - | 276.87 | 276.87 |
| | Edge Computing | Our | 27.38 | 1054 | 7.77 | **1089.15** |
| | | ResNet-50 | - | - | 1965.81 | 1965.81 |
| Folio (4160 × 3120) | Workstation | Our | 104.27 | 848 | 4.30 | **956.57** |
| | | ResNet-50 | - | - | 2073.29 | 2073.29 |
| | Edge Computing | Our | 223 | 8622 | 8.28 | **8853.28** |
| | | ResNet-50 | - | - | 22080.98 | 22080.98 |

Table 3: Benchmark of Processing Times for PolygoNet and ResNet-50 on Server GPU and Jetson Orin Configurations across Various Datasets.

| Dataset | Pipeline | Server GPU | | | Jetson Orin | | |
|---|---|---|---|---|---|---|---|
| | | Extraction (ms) | Inference (ms) | Total (ms) | Extraction (ms) | Inference (ms) | Total (ms) |
| FashionMNIST (28 × 28) | Contours None | 0.32 | 1.31 | 1.63 | 2.28 | 12.87 | 15.14 |
| | Contours Simple | 0.15 | 1.29 | 1.44 | 1.32 | 10.33 | 11.65 |
| | Contours TC89 L1 | 0.14 | 1.21 | 1.35 | 1.28 | 8.25 | 9.53 |
| | Contours TC89 KCOS | 0.09 | 1.15 | **1.24** | 1.40 | 6.68 | **8.08** |
| | ResNet-50 | - | 17.06 | 17.06 | - | 116.25 | 116.25 |
| Flavia (1600 × 1200) | Contours None | 9.37 | 2.05 | 11.42 | 13.80 | 28.38 | 42.18 |
| | Contours Simple | 8.20 | 1.68 | 9.88 | 8.29 | 21.18 | 29.47 |
| | Contours TC89 L1 | 8.09 | 1.43 | 9.52 | 7.98 | 19.19 | 27.17 |
| | Contours TC89 KCOS | 7.71 | 1.42 | **9.13** | 8.34 | 18.14 | **26.48** |
| | ResNet-50 | - | 276.87 | 276.87 | - | 1965.81 | 1965.81 |
| Folio (4160 × 3120) | Contours None | 64.30 | 2.92 | 67.22 | 223.00 | 36.62 | 259.62 |
| | Contours Simple | 43.17 | 2.17 | 45.34 | 64.96 | 24.34 | 89.30 |
| | Contours TC89 L1 | 43.63 | 1.85 | 45.48 | 42.61 | 18.48 | **61.09** |
| | Contours TC89 KCOS | 42.61 | 1.81 | **44.42** | 72.68 | 19.75 | 92.43 |
| | ResNet-50 | - | 2073.29 | 2073.29 | - | 22080.98 | 22080.98 |

# 6 DISCUSSION

The experimental results highlight PolygoNet's effectiveness in resource-constrained environments. Across all datasets. PolygoNet consistently requires significantly fewer floating-point operations (FLOPs) compared to ResNet-50 while maintaining competitive performance metrics. For instance, on the Folio dataset, PolygoNet achieves an accuracy of 78% with just 8.66 million FLOPs, compared to ResNet-50's 86% accuracy at 21.47 billion FLOPs. This substantial reduction in computational demand makes PolygoNet particularly suitable for applications with limited computational resources, such as embedded devices like the NVIDIA Jetson Orin Nano. Although ResNet-50 slightly outperforms PolygoNet in terms of accuracy and F1-score in certain scenarios—most notably on FashionMNIST—PolygoNet offers a compelling balance between performance and com-

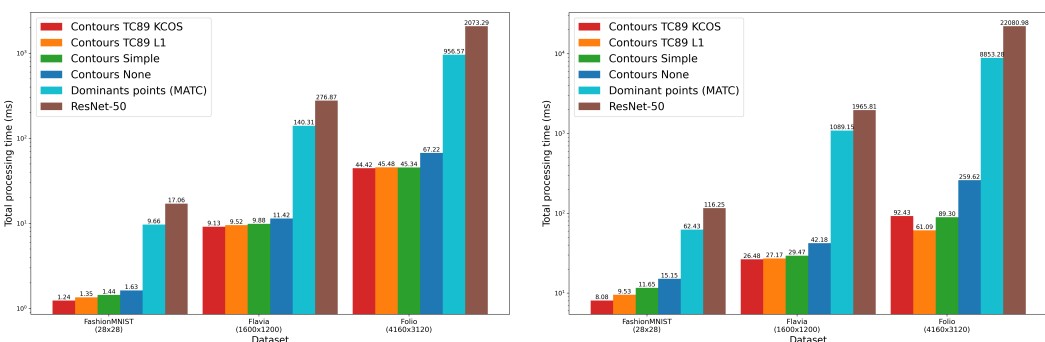

(a) Inference Time on *Server* Configuration    (b) Inference Time on *Jetson Orin* Configuration

Figure 4: Inference Time per Dataset and Approach on Different Configurations. (a) Comparison of processing time for various datasets using PolygoNet with four contour extraction methods and ResNet-50 on a *Server* setup. (b) The same comparison on a *Jetson Orin* embedded system. The y-axis is logarithmically scaled to highlight performance differences.

putational efficiency. On FashionMNIST, PolygoNet (Contours) attains an accuracy of 83% and an F1-score of 0.91, closely approaching ResNet-50's 90% accuracy and 0.93 F1-score, while operating with approximately 8.65 million FLOPs compared to ResNet-50's 80.38 million FLOPs. PolygoNet's advantage is further emphasized in embedded system configurations. On the Jetson Orin Nano, PolygoNet significantly outperforms ResNet-50 in processing time across all datasets, demonstrating its suitability for environments where speed and energy efficiency are critical. For example, on the Flavia dataset, PolygoNet (Contours TC89 KCOS) completes processing in 26.48 ms on Jetson Orin, compared to ResNet-50's 1,965.81 ms. The utilization of contour points in PolygoNet introduces slight performance enhancements over dominant points. On FashionMNIST, incorporating contours increases accuracy from 79% to 83% and the F1-score from 0.90 to 0.91. Additionally, the contour-based approach eliminates the need for the computationally intensive MATC (Modified Adaptive Tangential Cover) method used in extracting dominant points, thereby reducing processing time. Direct contour extraction not only preserves essential structural information such as shapes and object boundaries but also streamlines the inference process, resulting in faster and more efficient computations. However, these improvements come with a marginal increase in model complexity. For example, on FashionMNIST, the FLOPs increase from 8.52 million with dominant points to 8.65 million with contours. Despite this slight rise, the benefits in accuracy and processing speed justify the trade-off, making the contour-based approach a viable option even in highly resource-limited settings.

## 7    CONCLUSION

In this paper, we introduced PolygoNet, a new approach that utilizes polygonal contours and dominant points for efficient image classification with deep neural networks. By transforming input images into compact polygon representations, PolygoNet significantly reduces computational complexity, making it ideal for real-time and resource-constrained environments. Our experiments on benchmark datasets demonstrate that PolygoNet achieves competitive accuracy and F1-scores comparable to ResNet-50, while requiring a fraction of the computational resources. The integration of contour-based methods enhances PolygoNet's ability to capture essential geometric features, further improving classification performance without substantial increases in computational load. This tradeoff between accuracy and efficiency underscores PolygoNet's suitability for applications in edge computing and mobile AI. Techniques such as active contours (Marcos et al., 2018) and Bézier curves (Splines) can be used to encode contours for the pipeline. For more complex scenarios, models such as the SAMs (Kirillov et al., 2023; Ravi et al., 2024) can be employed to generate contours from predicted masks, despite their higher computational cost, but this remains to be explored. This approach would allow PolygoNet to be applied to more complex datasets and diverse real-world scenarios.

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
