# OpenReview forum: "PolygoNet: Leveraging Simplified Polygonal Representation for Effective Shape Classification"
_ICLR.cc/2025/Conference — ICLR 2025 Conference Withdrawn Submission_

### Official Review · Reviewer_gdR6 · 2024-10-19

**Soundness:** 1
**Presentation:** 2
**Contribution:** 1
**Rating:** 3
**Confidence:** 4

**Summary:**

The work proposes a contour and dominant point-based image classification model. By only considering the contour and dominant point, the work can speed up the training and inference speed of the neural network.

**Strengths:**

1 Fast inference speed.

**Weaknesses:**

1 In lines 68-72, the work mentions, 'This methodology aligns with cognitive processes observed in human visual perception, where recognition is often based on key structural features rather than exhaustive pixel-by-pixel analysis Biederman (1987); Koffka (2013).' This motivation/insight oversimplifies the image recognition task and human perception. Human perception and cognition systems are much more complex than this. Biederman (1987)'s work addresses the basic level of the cognition process. Human cognition operates on multiple levels and involves lateralization, where both hemispheres of the brain contribute to different aspects of recognition—such as broad categorization and fine-grained identification. This multiple-level, lateralized system integrates structural, contextual, and surface information, which goes far beyond the simplified structural approach suggested in the work. I encourage the authors to take a look at this literature review [2].

By the way, there is an interesting game called 'Who's That Pokémon?' In this game, the player needs to guess the name of the Pokémon based on its contour, which is more challenging than using pixel-by-pixel information.

2 In lines 23-25 and lines 65-67, the work mentions that the proposed method can filter out the background noise. This is an interesting direction, as spurious correlations [1] often arise from the background. However, the proposed method relies on a thresholding mechanism to filter out irrelevant content. I do not believe this approach is robust, especially in scenes with diverse backgrounds. The qualitative results are all based on datasets with simple backgrounds. Using the segmentation model is one way as mentioned in the conclusion, but the work does not consider integrating the model, limiting the work.

3 Only considering contour and dominant points is risky and underestimates the complexity of image recognition/classification tasks. Many image classification tasks depend on more than just contour recognition. For example, how can the proposed method distinguish different kinds of balls, such as basketballs, soccer balls, ping-pong, volleyballs, and baseballs, without knowing the texture, color patterns, or surface details? How can the proposed method distinguish the brand of vehicles? The Flavia dataset used to demonstrate recognition is favorable to the approach, as the shape and contour of the leaves are different across categories. If the proposed method wants to prove the classification ability, considering the examples I mentioned is more convincing.

4 The work emphasizes inference speed and tests it on edge devices, which is a highly relevant real-world consideration. However, the work does not provide a real-world demo; instead, it is tested on an existing dataset offline. Deploying this in a real-world scenario is a challenge that is not addressed. For instance, real-world scenes often have complex backgrounds, include multiple objects, and feature objects in various poses. None of these real-world considerations are taken into account in the work.

5 The work may not work well for non-rigid objects and objects with various poses. Although the datasets contain non-rigid objects such as clothes, these objects are well-posed in the dataset.

6 The baseline is just ResNet50. Only adopting one baseline is not convincing.

[1] Kim, Younghyun, et al. "Bias-to-text: Debiasing unknown visual biases through language interpretation." arXiv preprint arXiv:2301.11104 (2023).

[2] Palmeri, Thomas J., and Isabel Gauthier. "Visual object understanding." Nature Reviews Neuroscience 5.4 (2004): 291-303.

**Questions:**

The proposed method is limited (see 2,3,4,5 above) and underestimates the task and system (see 1, 2, 3 above). Please clarify if I have a misunderstanding in the weakness section.

---

### Official Review · Reviewer_54uS · 2024-11-04

**Soundness:** 1
**Presentation:** 3
**Contribution:** 2
**Rating:** 3
**Confidence:** 4

**Summary:**

The paper proposes a pipeline to classify input images with lower computational requirements compared to full-scale CNNs, potentially suitable for edge devices. The pipeline consists in running classical contour-extraction algorithms, optionally applying the MATC approach to extract dominant points from the contours, and then feeding the extracted points (contour or dominant) to an ad-hoc classification network architecture composed of self-attention with positional encoding and 1D convolutions, to leverage global and local context. Experiments are carried out on FashionMNIST, Flavia and Folio datasets, which include objects on regular backgrounds, comparing the results of the proposed method to a ResNet baseline. The proposed method uses significantly less FLOPs compared to the baseline, which translates to significantly lower inference times. The approach using dominant points uses marginally less operations compared to using the full contours, but it requires significantly longer overall time, albeit still less than the baseline, due to the MATC algorithm. Both proposed approaches achieve F1 scores comparable with the baselines, but reduced accuracy.

**Strengths:**

- The paper is well written. In particular the first 5 pages are very well written, clear and easy to follow, including also an overview of the MATC algorithm
- The idea is definitely interesting: using only contours (or, more generally, keypoints) to do classification can lower the computational requirements, which is important for edge devices, and it is also going in the direction of human-inspired techniques compared to processing every input pixel using CNNs or ViTs
- The proposed algorithm is effective in significantly reducing the runtime and number of operations compared to the ResNet baseline

**Weaknesses:**

- Even though the paper is generally well written, the first half of the paper includes repetitions when describing the proposed pipeline. The network architecture description at page 6, instead, could benefit from a more descriptive figure.
- A good part of the paper is used to describe the MATC-based approach, only for it to be shown as slightly worse than the contour-based approach and with significantly higher runtime. Can the authors elaborate on the importance of this variant of the proposed pipeline? For example if they see use cases in which it can be preferred over the contour-based variant. Otherwise, the paper could also be reformatted to more effectively target the question "how much information is needed for classification", to which the answer could be that contours already provide a good amount of information, but dominant points carry less information and require more execution time. If targeting this question, additional experiments would be required (such as including internal contours/edges, colours, etc).
- The paper claims "enabling our pipeline to demonstrate consistent performance across diverse and challenging conditions", however the experiments are carried out on datasets with very simple conditions, such as regular backgrounds, which can significantly simplify the task of contour extraction, key to the proposed pipeline. Either more challenging datasets should be employed, or the claim should be relaxed, and also clarified in the introduction that the pipeline is only suitable to images which can be easily binarized.
- The experiments only include one baseline: ResNet. A milestone of vision architectures, ResNet is certainly important, but a bit outdated compared to more recent architectures. ViTs should be included (and they usually require more computation, as the authors also note, which could benefit the comparison to the proposed approach), as well as other architectures targeted to low-compute devices, such as YOLO. Without these baselines, it is difficult to assess the validity of the method, even in controlled conditions such as the ones offered by the chosen datasets.

**Questions:**

The proposed architecture uses self-attention, which can potentially increase computation times over simpler architectures. Did the authors experiment with simpler architectures? If so, the results could be interesting to include.
Did the authors experiment with PointNet-like architectures, since the input is composed of contour/dominant points?

---

### Official Review · Reviewer_Mi3N · 2024-11-05

**Soundness:** 2
**Presentation:** 3
**Contribution:** 2
**Rating:** 3
**Confidence:** 3

**Summary:**

The paper proposes an approach to leverage efficient polygonal representations of input images. Polygonal representation offers a concise and flexible depiction of images. The proposed method transforms input images into polygonal forms using either dominant points or coordinates of contours. The polygonal representation 1) substantially reduces the computational burden associated with processing large image datasets, 2) accelerates the training process, 3) conserves computational resources, 4) facilitates improved generalization of the trained models, and 5) is suitable for real-time applications and resource-constrained environments. The polygon forms are used to train deep neural networks. The proposed method achieves comparable performance to SOTA methods, mitigate overfitting, and produce lightweight models suitable for edge computing.

**Strengths:**

+ The proposed method achieved
1) multiple outcomes (listed in the summary), making it unique.
2) results comparable to SOTA methods on multiple datasets (FashionMNIST, Flavia, and Folio).
+ The proposed method achieves impressive inference time (both on the server and Jetson Orin configurations, as shown in Figure 4) and total time on both devices (workstation and edge computing, as shown in Table 2).
+ The paper is well-written and easy to understand/follow.

**Weaknesses:**

- The paper is limited by qualitative results (restricted to one or two samples). It is unclear if there are complex cases and how the model performs.
- Experiments are limited to small-sized datasets.
- Resnet-50 is a slightly old method and is the only method used to compare with PolygoNet (Contours and DP, as shown in Table 1). It appears more like a technical report than a comprehensive comparison with recent state-of-the-art methods in the problem domain.
- The margin of improvement in the results due to the proposed method is negligible (Table 1) on all three datasets. Though F1 Scores are comparable, the accuracy values are lower.
- The proposed method is limited to extracting geometric/shape features from 2D images while ignoring color or other 2D feature details.
- The paper does not provide an ablation study to understand the contributions of individual components within the proposed method.

**Questions:**

+ Consider updating the paper with the recent state-of-the-art models (refer to https://paperswithcode.com/sota/image-classification-on-fashion-mnist) results on fashionmnist dataset.
+ Please consider using larger-sized datasets for quantitative results.

---

### Official Review · Reviewer_9Zmn · 2024-11-08

**Soundness:** 2
**Presentation:** 2
**Contribution:** 1
**Rating:** 1
**Confidence:** 3

**Summary:**

The paper proposes a method for efficient image classification. The main insight is distilling images to two slim representations - either a contour or dominant points of this contour - and use it for classification, inducing minimal computational costs both during inference and training.

**Strengths:**

The paper is concise, and explains the idea in simple words
There might be some interesting insight in using a very concise representation

**Weaknesses:**

Unmotivated problem
Non noval solution
Limited applicability
Poor results
Inadequate evaluation

**Questions:**

The paper tackles the problem of single object classification in an image. This is a VERY populated and saturated field, with techniques improving two tenths of a point with niche tricks being published. More specifically, the paper tackles efficient computing for edge devices. Again, this is a field that started years ago with dozens of publications and even products already running on edge devices around the world.
All this work is completely ignore by the paper. Instead, the paper suggests a naive and hand tuned reduction approach, and compares to a vanila resnet-50. Even in this dated comparison, there is no clear merit for the proposed approach.
There are still many questions left unanswered unfortunately:
* How would self-learned features behave for the same computational cost?
* What other alternatives for compression are there?
* How does one tackle more complex scenes?
* How does the method fair against shuffle net, mobile net and their many followups?

---

### Note · Authors · 2024-11-22

I have read and agree with the venue's withdrawal policy on behalf of myself and my co-authors.